# The Epistemology of Bacterial Virulence Factor Characterization

**DOI:** 10.3390/microorganisms12071272

**Published:** 2024-06-22

**Authors:** Matthew Jackson, Susan Vineberg, Kevin R. Theis

**Affiliations:** 1Department of Biochemistry, Microbiology and Immunology, Wayne State University School of Medicine, Detroit, MI 48201, USA; 2Department of Philosophy, Wayne State University, Detroit, MI 48201, USA; susan.vineberg@wayne.edu

**Keywords:** bacterial virulence factors, evolution of virulence factors, bacterial pathogenesis, reasoning bias

## Abstract

The field of microbial pathogenesis seeks to identify the agents and mechanisms responsible for disease causation. Since Robert Koch introduced postulates that were used to guide the characterization of microbial pathogens, technological advances have substantially increased the capacity to rapidly identify a causative infectious agent. Research efforts currently focus on causation at the molecular level with a search for virulence factors (VFs) that contribute to different stages of the infectious process. We note that the quest to identify and characterize VFs sometimes lacks scientific rigor, and this suggests a need to examine the epistemology of VF characterization. We took this premise as an opportunity to explore the epistemology of VF characterization. In this perspective, we discuss how the characterization of various gene products that evolved to facilitate bacterial survival in the broader environment have potentially been prematurely mischaracterized as VFs that contribute to pathogenesis in the context of human biology. Examples of the reasoning that can affect misinterpretation, or at least a premature assignment of mechanistic causation, are provided. Our aim is to refine the categorization of VFs by emphasizing a broader biological view of their origin.

## 1. Importance

The characterization of bacterial virulence factors (VFs) in the context of human biology may be influenced by an anthropocentric bias, and therefore, efforts to develop therapeutic interventions of their effects in medicine may be misdirected. We propose a reassessment of the reasoning processes that are used to define the role that a VF has in bacterial pathogenesis and encourage researchers to consider that there can be multiple plausible explanations for a given observation. Guidelines that encourage the consideration of the role of natural selection in the evolution of a putative VF are provided.

## 2. Introduction

In 1890, Robert Koch postulated conditions that associate the presence of a particular infectious agent with disease [1]. These postulates helped direct the search for discrete virulence factors (VFs) that contribute to disease. Historically, the identification of a VF is largely based on empirical observations made in vitro. Namely, pathogenesis involves tissue death, and tissues are composed of somatic cells, so if we observe the killing of somatic cells by a VF in vitro, then the conclusion is that it must have a role in pathogenesis. We teach our microbiology students that to qualify as a pathogen, bacteria must colonize and evade host defenses, cause tissue damage, and promote their own dissemination. We follow-up by providing lists of VFs categorized according to their role in each step of this process. However, this may be an overly anthropocentric view that ignores the evolutionary history of microorganisms in general and overemphasizes the relationship between mammals, especially humans, and bacteria. This perspective disregards a fundamental aspect of Darwin’s theory of evolution via natural selection [2] by presuming that apparent VFs possessed by environmental bacteria evolved to infect mammals, wherein in reality, the collection of gene products designated as VFs have plausibly evolved to support the survival of bacteria in the broader environment, with the occupation of mammals’ bodies appearing later in their history as a new niche that may necessitate some degree of prior pathoadaptation [3]. Currently, nucleic acid amplification technologies (i.e., PCR and sequencing) may be used to identify potentially new VFs. Lacking any other experimental evidence, these approaches can misidentify VFs and thereby misdirect downstream research efforts. In this essay, we examine the epistemology of VF characterization and introduce criteria that can be used for categorization. Our goal is not to introduce extreme epistemic rigidity that would hamper the field but to encourage an unbiased reasoning process to be used for VF characterization.

A review of the literature reveals evolutionary biologists who have sought to define the environmental pressures that resulted in the development of bacterial factors that defend the bacterium against death by protists, and there are microbiologists from various fields who seek to delineate the roles that these factors play in disease along with their regulation and secretion [4,5,6,7,8]. Yet, it seems likely that what we regard as VFs may have a primary role in defending the bacterium against environmental pressures and that these factors may ultimately have served a secondary role for the bacterium when faced with colonizing new niches in mammalian hosts [9,10,11,12]. While the belief that the evolution of a VF as a means to win the arms race against predators has warrant, the reasoning used to infer their role in mammalian disease should undergo reappraisal. The redefinition of Koch’s postulates in the 20th century focused on the existence of a gene that produces a VF [13]. This approach serves to validate the role of a discrete gene product in pathogenesis. However, in vitro assays and animal models used in these experiments can never be wholly representative of the human disease process or the selective pressures that direct pathogenesis [14]. Furthermore, conclusions drawn from genomic analyses that are increasingly being used as a form of inductive reasoning by analogy in the quest for identifying new VFs may be overly hasty [15]. Without considering their evolutionary role as broader survival factors, the characterization of VFs is potentially subject to anthropocentric bias. A global approach to VF research will mitigate the negative impact that various biases can have on the field of microbial pathogenesis.

Families of putative VFs have been identified through genomic data mining based on the assumption that a particular gene product is responsible for a discrete step in the infectious disease process [15,16]. Therapeutic interventions may be investigated using these curated databases without a clear set of guidelines for defining the role, if any, of a particular VF in disease [17]. Novel therapeutic agents and vaccines have been developed using this approach. For example, a small molecule inhibitor of cholera toxin expression was isolated using high-throughput screening [18], and the yersinia type 3 secretion system effector protein was used to construct a hybrid vaccine that showed protection in a mouse model [19]. However, the exploration of therapeutic strategies aimed at a misidentified VF risks wasting research efforts. We propose the recognition of the roles that natural selection combined with pathoadaptation have in the evolution of VFs and to remain cognizant of the biases that can be introduced in hypothesis development. As expressed by Brown et al. [20] and attributed to Joshua Lederberg, “… the intellectual approach taken to study parasitism is perhaps most effective when an anthropocentric view is rejected in favour of an ecological view”. We thus propose a distinction between those gene products that provide offensive and defensive capabilities to the bacterium from those that provide general survival functions such as colonization and nutrient acquisition [21]. When the offensive and defensive capabilities extend to a lifecycle in the mammalian host, then the former would be described as VFs, while the latter would be designated fitness factors [22].

## 3. Origin and Evolution of Virulence Factors

The concept of a VF as a discrete gene product which directly contributes to the ability of a bacterium to cause disease accommodates our heuristic tendencies for narrative explanations of causation. As the concept that bacterial pathogens are defined by their list of VFs, a search for new genes has been supported by the creation of nucleic acid databases based on sequence homologies [15,23]. These efforts can potentially lead to an exponential increase in the number of sequences identified as VFs in species previously considered as commensals or inhabitants of various niches in the extracorporeal environment [16]. The realization that bacteria and viruses developed a Darwinian relationship with protists eons before the appearance of mammals should compel us to consider from an evolutionary perspective the question of why bacteria produce VFs. Evolution did not consider the ultimate effect on mammals when generating gene products that allow the earliest life forms to colonize and evade the defensive response of single-cell eukaryotes [24]. Bacterial pathogenesis and the identification of VFs began as a field of investigation with the discoveries of van Leeuwenhoek, Pasteur, and Koch leading to diagnostic approaches and therapies based on sound reasoning [25]. Admittedly, an anthropocentric bias in bacterial pathogenesis has enabled bold predictions facilitating advances such as vaccine development that have served to mitigate the impacts of infectious disease. However, searches based purely on sequence homologies are prone to confirmation bias combined with limited empirical evidence that can ultimately lead to misdirected efforts to design therapeutic interventions.

Evolutionary biology research has revealed a diverse group of bacterial pathogens that have a protist host which most likely served as a training ground for their adaptive survival and supported the development of the nefarious traits that contribute to disease [26,27,28,29]. A prokaryote’s capacity to evade protist grazing most likely enabled it to evade a cell-mediated immune response in a mammalian host. While these pathoadaptive characteristics provide a general survival capacity, the unique human host specificity that we see with some VFs such as adhesins and the binding subunits of exotoxins seems to argue against the protist training ground hypothesis. Did the receptor binding specificity of an adhesin evolve after bacteria encountered a new mammalian niche, or did they exist prior to that encounter—were they required for binding to a nonmammalian receptor? There is also a question of fitness costs to the bacterium when it needs to express a survival factor. Bacteriophages with their capacity for horizontal gene transfer (HGT) may have evolved as prokaryotic symbionts serving as reservoirs for the rapid deployment of the genes needed for expansion into new niches thereby obviating the need for bacteria to constitutively expend unnecessary energy [30,31,32,33,34]. Pathoadaptive mutations in the bacterial genome that result in gain- or loss-of-function phenotypic changes, either by mutation or HGT, permit a commensal bacterium to colonize a new environment, acquire nutrients, and when necessary, evade the host immune response [3,35]. The concomitant tissue damage and inflammatory response are what we recognize as disease. A gain-of-function by mutation example such as *Pseudomonas aeruginosa* alginate production during lung infections [36,37] and gain-of-function by HGT example such as the acquisition of the Shiga toxin-converting phage by *E. coli* [30] illustrate the concept of heritable pathoadaptive mutations providing survival benefit to a bacterial population in a new niche. These phenotypic changes which aid survival and are heritable within the bacterial population can be considered examples of Lamarckian evolution within a Darwinian framework [15,38,39].

## 4. Ecological Arguments Supporting Non-Anthropocentric Roles of Bacterial Virulence Factors

We can no longer think in absolute terms of pure pathogen and pathogen-driven disease but must instead consider the concept of balanced pathogenicity in a damage-response framework [40]. A critical next step in this process is to reformulate the role of a VF fit within this model. We should view VFs as factors that serve a survival function for the bacterium in a specific ecologic niche such as resisting the killing processes inside the endocytotic vesicle of *Acanthamoeba* or a human macrophage [5,6,7,26,27,37]. Host response dictates whether the bacterium will persist as a quiescent intracellular life form or escape from its phospholipid enclosure using a cytolytic exotoxin. Although there may be VFs that possess the capacity to cause tissue damage irrespective of the host (e.g., the clostridial neurotoxins [41]), the damage-response framework model informs us that commensalism is relative and that disease causation may be considered an inappropriate response by the mammalian host to the bacterium. It may be possible one day to accommodate therapeutic approaches that take into consideration an individual host’s immune response to infection.

If a bacterial pathogen is understood as the product of eons of natural selection in which it developed co-resident relationships with and defenses against both bacteriophage and bacterivorous protists [22], then there should be more experimental models that explore the host–parasite relationship at its source using, for example, *Acanthamoeba* or *Tetrahymena*. It is important to maintain the perspective that bacteria interact with a variety of organisms including protists, fungi, mammals, and other prokaryotes. Kebbi-Beghdadi and Greub [42] propose using protists as eukaryotic hosts to isolate pathogens from environmental samples during an outbreak or from clinical specimens when traditional culture methods fail. Amoeba may be used as a model system that more closely mimics bacterial processes at the level of a mammalian cell. The relationship of *Legionella pneumophila* inhabiting *Acanthamoeba castellanii* has been well characterized [26], but there is limited information on other bacterial pathogens and their protist hosts despite the demonstration that these relationships are seemingly widespread [5,8]. Virulence attributes such as biofilm formation, quorum sensing, exotoxins, and siderophore production which have a role in bacterial survival when encountering their protist predator could also conceivably play a role in the mammalian host [6,22,36]. The capacity for molecular inhibitors to protect a protist host from a VF produced by a co-resident bacterium could be assessed prior to tissue culture and animal assays. Vertebrate cell culture and animal host models represent a relationship that evolved several billion years later and are used as a proxy for human infection. It would be naïve to indiscriminately criticize the role that traditional models have had without recognizing their substantial contributions to combatting infectious disease. Nevertheless, we should acknowledge that the extrapolation of these experimental results to human disease is subject to heuristic biases [43]. Rigorous science should always be aware of bias, and authors should explicitly state the limitations of a particular study. Two examples of established bacterial pathogens, *Staphylococcus aureus* and *Pseudomonas aeruginosa*, will be used to illustrate the processes for characterizing specific VFs. The rationale for choosing these two bacteria is based on the number and the variety of putative VFs that have been described for each of them. Arguments applied to these disease-causing bacteria may be applied to other human and zoonotic pathogens as well as to plant pathogens.

## 5. Two Case Studies: *Staphylococcus aureus* and *Pseudomonas aeruginosa*

Table 1 describes putative VFs that have been isolated from two bacterial pathogens, *S. aureus* and *P. aeruginosa*. VFs such as *S. aureus* alpha toxin and *P. aeruginosa* alginate and elastase (LasA) were excluded from the table because their role in human disease has been established by criteria that surpass those cited in Table 1 [44,45]. Table 1 is not presented to imply that research characterizing the bacterial pathogens *S. aureus* and *P. aeruginosa* has been entirely misdirected. Investigations of established and putative VFs that use available methods and extrapolation from in vitro and in vivo analyses to humans is currently the most appropriate approach to advance our understanding of microbial pathogenesis. Scientific discovery that is based on current knowledge and inductive analogy can be the best approach to developing testable hypotheses. The process of describing VFs is subject to underdetermination, which is the concept that the information currently available to researchers at a given time is insufficient to define what beliefs should be held in response to that information [46]. With time, methods of discovery improve, seminal discoveries are made, and the experimental data improve, which ultimately advances the field. The purpose of this essay is to encourage vigilance with respect to the reasoning process and to support the use of strong inference [47]. Strong inference promotes clarity in hypothesis development, which, along with a considerable amount of empirical information, allows for the formulation of multiple sharp hypotheses. Strong inference involves proposing definitive alternative hypotheses, which may be tested and compared. In the best case, all but one can be eliminated (with high probability), thus cutting down on the possibilities and raising the probability of the remaining hypotheses. For example, *S. aureus* produces a number of hemolysins that presumably contribute to tissue damage and disease, making them attractive targets for therapeutic intervention. Strong inference can be applied in this example by testing the alternative hypothesis that a particular hemolysin evolved solely for nutrient acquisition in the host and its role in pathogenesis is inconsequential. The role of this putative VF in disease can be granted warrant by establishing empirical evidence supporting or refuting the alternative hypotheses.

### 5.1. Staphylococcus aureus

*S. aureus* deserves consideration in the discussion of VFs. It is a commensal that colonizes the anterior nares of up to 27% of the human population and has the capacity to express proteins that enable it to colonize wounds and prosthetic devices while producing a variety of exotoxins that can destroy tissue and superantigens that induce anergy [44,78]. Genes producing these VFs that seem extraordinarily adapted to the human host are carried by mobile genetic elements (MGEs) that can provide a bacterial host with the survival tools that it needs: factors for colonization, the alteration of the antigenic surface structure, and toxins [79]. *S. aureus* has a versatile selection from its mobile libraries including phage, genomic islands, and pathogenicity islands that carry genes for many of its well-studied VFs such as enterotoxin A, Panton-Valentine leukocidin, and toxic shock syndrome toxin-1 [80], as well as plasmids and transposons that distribute antibiotic resistance genes. *S. aureus* presents itself as a valuable model for bacterial pathogenesis and the role of VFs in disease because the evolution of ancient staphylococci into the less virulent coagulase negative staphylococci and *S. aureus* was a result of the acquisition of genomic islands by the latter species [81]. The agr system is a quorum sensing system that controls VF gene expression and is believed to be responsible for the temporal expression of those factors that mediate colonization followed by dissemination through the tissue [82]. However, in vitro studies and epidemiologic analyses fail to fully support a consistent role of the agr system in pathogenesis [83,84]. The expression of the agr system is an example of an adaptive response that enables the persistent colonization of the human host [85]. The evolution of a regulatory system such as agr and functional redundancy that enable the long-term colonization of a particular niche can confound the use of mutated bacterial strains for the investigation of pathogenesis due to the role that many putative VFs have in survival [44]. Loss-of-function mutations in degradative enzyme gene sequences that are intended to mitigate tissue destruction in an animal model will impact pathogen fitness by impeding nutrient acquisition and perhaps immune evasion. Therefore, the characterization of functionally redundant exoenzymes and regulatory sequences as VFs should be subjected to additional criteria that establish their roles in pathogenesis.

*S. aureus* predominates as a human colonizer because it can gain resistance to multiple antibiotics, with resistance to methicillin and vancomycin being the most significant threats to human health [86]. Carrying the methicillin resistance cassette has significant fitness costs for the bacterium, but its existence on a plasmid supports the maintenance of the population because cheaters are transformed in the event of a selective challenge [80,87]. An important consideration is whether the reason for the genesis of these resistance genes is poor antibiotic stewardship since the middle part of the 20th century or whether this is an example of confusing correlation with causation. Evolutionary history does not support the resistance gene genesis argument since the appearance of the genes predates the use of methicillin [88]. Antibiotic resistance genes may have evolved to resist other environmental challenges, and we became aware of their existence with the advent of these therapeutic agents [89]. Therefore, overuse may be blamed for their dissemination via MGE but does not explain the evolution of these genes [90,91]. Some of the more virulent strains of *S. aureus* isolated from severe cases of bacteremia and pneumonia have been shown to not produce the exotoxins classically associated with this pathogen [83] which is contrary to the narrative that these proteins are required as part of the disease process. Evolution studies have demonstrated loss-of-function due to nonsense mutations in the AraC-family, which is a transcriptional regulator of stress response, as well as the impaired function of the agr global regulator [78]. This suggests there may be a fitness tradeoff for an opportunist such as *S. aureus* with some gene products lost or transcriptionally silenced during the transition from commensalism to pathogenesis.

The role that biofilms have in the survival of bacteria in their natural habitats was initially revealed by microbial ecologists and was subsequently followed by the application of these findings to human health [5,6,10,33,92,93,94,95,96]. These discoveries resulted in numerous reports describing the role of biofilms in dental disease as well as in iatrogenic infections [97]. The realization that bacteria possess the capacity to form communities that use quorum sensing for cell–cell communication has contributed to a deeper understanding of the relationship between microbes and their hosts [92]. One hypothesis is that microbial adaptation to the extracorporeal environment, which provided a survival advantage by resisting bacterivorous protist predation and chemical assault by other bacteria, aided the colonization of the human oral cavity—a niche that can yield a rich source of nutrients [96,98]. It would similarly follow that a colony of staphylococcus that was inadvertently relocated from the hands of a surgical team to a prosthetic device may switch to biofilm formation to resist cell-mediated and humoral defenses [44,78]. The hypothesis would be that all strains of staphylococcus that colonize the skin possess the *potential* to engage in biofilm formation when faced with a new challenge to their existence. Would failure to demonstrate the existence of a quorum sensing operon in all strains of *S. aureus* suffice as the falsification of this hypothesis?

### 5.2. Pseudomonas aeruginosa

*Pseudomonas aeruginosa* in the lung of a cystic fibrosis patient is one of the most striking examples of adaptation to host colonization [96]. Considered an opportunistic infection, *P. aeruginosa* undergoes a phenotypic conversion to biofilm formation which allows it to resist the innate, cell-mediated, and humoral defenses in the lung [37]. Its capacity to cause a persistent infection can be attributed to adaptation in the broader environment wherein *P. aeruginosa* has been shown to use quorum sensing to coordinate a mob attack on amoeba [7]. In addition to biofilm formation, *P. aeruginosa* has a variety of other attributes which presumably contribute to its pathogenesis. Its siderophores and type 3 secretion system serve as environmental defenses against protists as well as competing bacteria [36,99]. *P. aeruginosa* produces exotoxins which clearly fit the working criterion for a VF and can serve as an exemplar for the VF characterization discussion. As an example, exotoxin A (ExoA) is a discrete gene product with the archetypal AB subunit structure of a bacterial toxin. It is an ADP-ribosylating enzyme that selectively modifies mammalian elongation factor 2 [33]. ExoA works superbly in vitro and has contributed to our understanding of eukaryotic protein synthesis [66]. Another key characteristic is that the pseudomonas exotoxin A has the identical mechanism of action as the diphtheria toxin (DT) A subunit. DT, which is produced by the respiratory tract pathogen *Corynebacterium diphtheriae*, causes a systemic disease considered a paradigm for bacterial toxinosis [40]. Biochemical cross-linking studies and sequence analysis revealed that the ExoA and DT A subunits share the same active site amino acids in a conserved pocket [100]. The role of diphtheria toxin in disease has been established by the overwhelming success of the DT toxoid vaccine, which reduced the incidence of an infectious disease that had a significant impact on the human lifespan [25,101]. The functional equivalence of the ExoA and DT A subunits are apparent, and both are produced by bacteria with established roles in pathogenesis, which may potentially lead to the error of unwarranted extrapolation from the conclusive findings associated with DT A to ExoA (Table 1). In support of the hypothesis that ExoA has a role in human disease is a 1979 study by Pollack et al. [67] that demonstrated increased antibody production to ExoA and lipopolysaccharide in patients who survived pseudomonas infections, satisfying our proposed guideline for an immunologic response to infection by a VF-producing pathogen (point 7, below). Bacterial exotoxins such as ExoA deserve special consideration as VFs because, when considered as distinct gene products, they seem to be well-adapted tools that contribute to tissue damage and, in some instances, promote the dissemination of their bacterial producer [102].

Exotoxins are divided into membrane-disrupting cytolysins and cytotoxins that possess specialized enzymatic activity that selectively alter eukaryotic cell function [103]. Based on their mechanisms of action, we can speculate that cytolysins have defensive capabilities resisting bacterivorous predators or serve the bacterium by facilitating escape from an endocytotic vesicle into the sheltered environment of the cytoplasm [104]. So, we may predict that the ubiquitous phospholipid bilayer that comprises cytoplasmic membranes is a universal target for cytolysins which serve to defend the bacterium against a protist in the environment or a macrophage when infecting a mammalian host. On the other hand, cytotoxins seem to have evolved specific targeting capabilities with the enzymatic activity of the cytotoxin A subunit uniquely adapted to specific cellular functions. In some examples, such as the calmodulin requirement of pertussis toxin, the enzyme can subvert eukaryotic signaling mechanisms to affect increased cytoplasmic cAMP concentrations [105]. Therefore, it would appear that cytotoxins are the prototypic bacterial VF that exist to cause disease. Their functions include the hallmarks of bacterial pathogenesis, which are the evasion of the host immune response, tissue damage, and the dissemination of the pathogen. However, this hypothesis neglects the Trojan horse model that has been proposed to characterize the microbe and predator arms race [8,34]. Toxin A subunits, with their enzyme function devoid of the receptor-binding B subunit, delivered directly into the cytoplasm of a protist will kill it [41]. So, if toxin A subunits evolved to kill the predator when the bacterium has been ingested, what were the selective pressures that drove association with a B subunit with highly specific receptor recognition? Exotoxin B subunits are specific to a particular cell receptor, and in some examples, such as cholera toxin, pertussis toxin, and the Shiga toxins, they are structurally complex [105]. Their closest analog in the prokaryotic world are pore-forming cytolysins which bind to specific membrane receptors, penetrate, and eventually disrupt the phospholipid bilayer. To form one of the Shiga toxins, for example, the C-terminus of the A subunit penetrates the B pentamer pore and is “locked” in place through electrostatic interaction [106]. Similar complexity can be observed with the type 3 secretion apparatus of bacterial pathogens which serves specific functions for transmembrane effector delivery [33,107]. The origins of the cytotoxin B subunit and the evolutionary pressure that led to their association with the A subunit are worthy of exploration.

## 6. Sources of Error in the Characterization of Virulence Factors

A current paradigm is that bacterial pathogenesis is associated with the variety and potency of VFs that allow the microbe to survive the mammalian host as a new niche and, depending on the appropriateness of the host response, cause disease [108]. The discovery and characterization of bacterial toxins date back to the characterization of anthrax by Pasteur and have led to the development of attenuated toxoids to prevent human diseases such as diphtheria and tetanus [25]. This is a heuristic process that involves projecting a known onto an unknown property as a categorical induction, which is a powerful process for discovery [43]. This type of induction involves projecting a known property of one item in a particular category onto another item in the same category. Genomic analysis is a powerful method to achieve this task and should be used to identify putative VFs—even from bacteria not known to have a role in disease that may have evolutionary links to well-known pathogens [15]. However, investigators should remain continuously aware of the variety of cognitive biases that can contribute to scientific error. In this example, based solely on in vitro empiricism, a protein may be mis-categorized as a VF that contributes to disease when its only purpose is to defend a bacterium from protist grazing. A comparison of the gene sequences of a putative VF that is expressed in both protist and mammalian hosts could lend insight into key pathoadaptive changes. Conventional thinking can act as an obstacle to the development of better models that consider the capacity for a bacterium to adapt to a new ecological niche followed by an investigation into their role in pathogenesis [109].

In biologic systems, the proof of causation is contingent upon empirical observation, and, in the quest for VF identification, we want a precise characterization of the sort of observations that are required to determine disease causation. As described by Fredericks and Rhelman [13] in the context of Koch’s postulates, the introduction of new technologies in the field of microbial pathogenesis results in an ongoing need to reassess how virulence attributes are defined. In his review addressing causation and disease, Evans [110] points out that technological developments in the 1960s–1970s yielded the identification of new agents (e.g., viruses) by electron microscopy which had no known role in disease at the time. Because these viruses could not be propagated in tissue culture, serological analysis was used to establish immunological proofs of causation. Therefore, we have included immunologic response as a criterion for the identification of VFs in our guidelines (see below). Sequencing technologies have the advantage of detecting organisms that are not culturable or observable by other existing technologies. However, while sequence homologies are a valuable first step, they cannot be the sole criterion used for the identification of new VFs. Additional evidence is required such as the correlation of the disease process with gene mutation and complementation (using the best available model), the detection of an immune response following natural infection, and demonstration that vaccines or target therapeutic agents ameliorate disease [21].

Should we pause the quest to identify new bacterial VFs by searching for sequence homologies to consider their epistemological origins? This question was raised over two decades ago [111] and remains a current concern. We propose considering the biases that can impact the growth of scientific knowledge [112] in the context of VF identification. For example, if a gene product is shown to have a cytopathic effect in vitro, then it becomes labeled as an exotoxin with a presumed role in pathogenesis. This is a potential example of heuristic weakness, which is the need to categorize empirical observations to fit a familiar narrative [113]. Notwithstanding the number of genes identified as VFs by sequence homology, we should strive to identify what would *disprove* their role in pathogenesis. We suggest that a cautious approach to the classification of a putative VF should be exercised. For example, what is the evolutionary origin of the gene, and are there evolutionary pressures to maintain the gene? Is the gene for the putative VF carried on an MGE and transmitted amongst a bacterial population by HGT, or is it a relatively stable element as seen in obligate intracellular pathogens such as *Chlamydia trachomatis* and *Mycobacterium tuberculosis* [17]? Are global regulators that are resident in the bacterial chromosome responsible for controlling the expression of the putative VF, or are the regulatory genes carried on a pathogenicity island supporting the hypothesis that selective pressure maintains these elements as a discrete unit? We need to establish that if a criterion for the identification of a VF is to ultimately develop a strategy to ameliorate its effect during an infection, then that should obviate gene products that are constitutively expressed and serve as fitness factors as described by Kreibich and Brussow [21,22]. The goal of developing our guidelines was to restrict VF characterization to those factors that have been shown to have a direct role in pathogenesis [108], bearing in mind the processes of both biological and epistemological evolution that have affected this effort in the past.

## 7. Proposed Guidelines for Characterizing Virulence Factors

In an effort to avoid bias and to foster scientific rigor, we submit the following considerations when characterizing a bacterial gene product as a VF:If the identification of a VF is based entirely on in vitro studies with activity extrapolated to a role in mammalian pathogenesis, then it should be designated as a “putative VF” until evidence provides sufficient warrant for categorization.If the identification of a VF is based on sequence homology without fulfilling significant criteria establishing its role in disease, then it should be designated as a “putative VF” until evidence provides sufficient warrant for categorization.Fitness factors and host interaction factors that do not extend the disease-causing potential beyond commensalism should not be categorized as VFs.

Necessary conditions for VF categorization:
4.The VF is produced by a bacterium with an established role in disease.5.An evolutionary role for the VF in niche survival has been correlated to disease causation.6.A role in pathogenesis has been empirically established by loss-of-function (mutation or suppression) and gain-of-function (complementation) experiments.7.An immunologic response is observed in response to infection by the VF-producing pathogen.8.Therapeutic strategies such as active or passive immunization to the VF or a targeted therapeutic agent have been shown to prevent disease progression.

## 8. Conclusions

The guidelines for VF characterization are intended to encourage the rigorous appraisal of the evidence that purportedly supports the factor’s role in disease causation. One goal of this essay is to propose the consideration of the role that VFs have in a broader evolutionary context beyond mammalian disease. One proposed approach would be to use their natural protist hosts as eukaryotic models of pathogenesis [42] in conjunction with the traditional use of mammalian-derived tissue culture assays. We seek to broaden the scope of VF characterization to include evidence that provides a definitive link to human disease prior to investigating therapeutic interventions [114]. A concept not made explicit in our guidelines is the distinction between the activities of a VF at the molecular level and pathogenic potential [108]. In that context, it would be necessary to consider the potential for damage caused by bacterial VFs irrespective of the host response.

## Figures and Tables

**Table 1 microorganisms-12-01272-t001:** Putative VFs of *Staphylococcus aureus* and *Pseudomonas aeruginosa*.

VF (Gene Designation)	VFDB Categorization ^a^	Findings and Limitations	Reasoning Error ^b^	Reference
*Staphylococcus aureus*				
Beta hemolysin (*hlb*)	Membrane-acting exotoxin	Cytotoxic Sequence homology with sphingomyelinase	False premise Unwarranted extrapolation	[48]
Delta hemolysin (*hld*)	Membrane-acting exotoxin	Cytotoxic Regulation of agr operon	False premise	[48]
Panton-Valentine leukocidin (*lukS-PV*, *lukF-PV*)	Membrane-acting exotoxin	Cytotoxic Not produced by all strains	False premise Unwarranted extrapolation	[49,50,51,52]
Phenol-soluble modulin-alpha peptides	Membrane-acting exotoxin (*psmα1*, *psmα2*, *psmα3*, *psmα4*)	Cytotoxic Limited host range in animal models	False premise Unwarranted extrapolation	[49,50]
*S. aureus* binder of IgG (*spa*)	Immune modulation	Immunoglobulin binding Sequence homology with protein A	False premise Unwarranted extrapolation	[53]
Nuclease (*nuc*)	Immune modulation	Digestion of neutrophil extracellular traps	False premise	[54]
Fibrinogen-binding protein (*fnbA*, *clfA*)	Adherence	Hemagglutination, receptor binding Antiphagocytic activity	False premise	[55,56,57]
Coagulase (*coa*)	Exoenzyme	Hemagglutination and antiphagocytic Immunization provided protection in mouse model	False premise Unwarranted extrapolation	[58,59]
Staphylokinase (*sak*)	Exoenzyme	Hemagglutination and antiphagocytic Anticoagulant treatment provided protection in mouse model	False premise Unwarranted extrapolation	[59,60]
Capsule (*cap* operons)	Immune modulation	Antiphagocytic Not produced by all strains	False premise Unwarranted extrapolation	[44,61,62]
PIA/PNAG (*icaADBC*)	Immune modulation	Antiphagocytic biofilm component Not produced by all strains	False premise Unwarranted extrapolation	[44]
Staphylococcal complement inhibitor (*scn*)	Immune modulation	Interference with complement	False premise	[57,63]
*Pseudomonas aeruginosa*				
LasB (*lasB*)	Effector delivery system	Immune evasion via modification of surface proteins	False premise	[64]
Phospholipase C (*plcH*)	Exotoxin	Degradation of lung surfactant in murine cystic fibrosis model	False premise	[65]
Exotoxin A (*toxA*)	Exotoxin	Antibody response during infection Functionally analogous to DTA ^c^	Unwarranted extrapolation	[66,67]
Exo S,T,U,Y (*exoSTUY*)	Effector delivery system	Functionally analogous to CyaA and EF ^d^	Unwarranted extrapolation	[68,69,70]
Lipase A (*lipA*)	Biofilm formation	Antiphagocytic biofilm component	False premise	[71]
Alkaline protease (*aprA*)	Exoenzyme	Immune evasion via modification of surface proteins	False premise	[64]
Type IV pili (*pilA-D, F-I*)	Adherence	Receptor binding in vitro	False premise	[72]
Protease IV (*prpL*)	Exoenzyme	Immune evasion via degradation of immunoglobulin, complement, and interleukin 22	False premise	[73,74]
Pyocyanin (*phz* operons)	Nutritional/Metabolic factor	Environmental survival Suppresses immune response	False premise	[75]
Pyoverdine/Pyochelin (*pvd*, *pch* operons)	Nutritional/Metabolic factor	Provides competitive advantage over Aspergillus in cystic fibrosis lung	False premise	[76,77]

^a^ VFDB categorization: http://www.mgc.ac.cn/VFs/VFcategory.htm (accessed on 28 May 2024). ^b^ False premise: if P, then Q and P; therefore, Q is a logical syllogism, but the validity depends on a true premise (P). In this case, the premise that if a VF kills cells in vitro or causes disease in a mouse model, then it causes disease in all animals (e.g., humans) may not be warranted. Unwarranted extrapolation: the categorization of a protein as a VF based on sequence homology or by an analogous mechanism to an established VF does not have warrant without additional proof. ^c^ DTA: *Corynebacterium diphtheriae* toxin A subunit. ^d^ CyaA: *Bordetella pertussis* adenylate cyclase toxin; EF: *Bacillus anthracis* anthrax toxin edema factor.

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
