# Peer review of "The Epistemology of Bacterial Virulence Factor Characterization"

_microorganisms, 2024, doi:10.3390/microorganisms12071272_

Round 1

Reviewer 1 Report

Comments and Suggestions for Authors

The authors present an intriguing argument for redefining what constitutes a virulence factor, focusing away from functions that simply allow the microbe to survive in different environments. They present two in-depth examples, and I would also add there are many examples of similar pathogens from the group of common foodborne pathogens, such as Listeria monocytogenes. 

My main comment is regarding not including plant pathogens - the focus here is on what we would commonly think of as human pathogens. do the authors think the same definitions should be applied for plant (and zoonotic) pathogens as well? would the definitions change at all?

For table 1 - bacterial names should be italicized in the title. I also think it would be helpful to include gene names or ORF IDs for the virulence factors listed.

Author Response

We are greatly appreciative of the reviewers and the editor for their suggestions on how to improve the manuscript.

Reviewer comments are in black font and our responses to those comments, as well as changes to the text, are in blue font. Page and line numbers of amended text reflect locations within the tracked changes version of the revised manuscript.

Reviewer 1

Comment #1

  1. The authors present an intriguing argument for redefining what constitutes a virulence factor, focusing away from functions that simply allow the microbe to survive in different environments. They present two in-depth examples, and I would also add there are many examples of similar pathogens from the group of common foodborne pathogens, such as Listeria monocytogenes.
  2. Response: The authors agree with the reviewer’s comment that the arguments presented in this manuscript may be applied to foodborne pathogens such as Listeria monocytogenes as well as extending beyond human pathogens to zoonotic and plant pathogens. The two examples used in the review, Staphylococcus aureus and Pseudomonas aeruginosa, were chosen due to the sheer number of extracellular factors that have been labelled as virulence factors and in the authors’ opinion, without sound reasoning.
  3. Lines amended in the document: page 1, lines 180-182

D: Amended text: Arguments applied to these disease-causing bacteria may be applied to other human and zoonotic pathogens as well as to plant pathogens. 

Comment #2

  1. My main comment is regarding not including plant pathogens - the focus here is on what we would commonly think of as human pathogens. do the authors think the same definitions should be applied for plant (and zoonotic) pathogens as well? would the definitions change at all?
  2. Response: The authors agree that the examples used in the manuscript are limited to two human pathogens, S. aureus and P. aeruginosa, but could be expanded to include zoonotic and plant bacterial pathogens. The two examples in the manuscript were chosen to illustrate research conclusions based on the potentially false premise that extracellular factors that show in vitro effects can be confidently extrapolated to human disease. A goal of this manuscript is to stimulate discussion on contemporary perspectives on bacterial virulence factors amongst experts in the field. This reviewer’s comments serve to stimulate future discussion that could be highly valuable to the field.
  3. Lines amended in the document: page 1, lines 180-182

D: Amended text: Arguments applied to these disease-causing bacteria may be applied to other human and zoonotic pathogens as well as to plant pathogens. 

Comment #3

  1. For table 1 - bacterial names should be italicized in the title. I also think it would be helpful to include gene names or ORF IDs for the virulence factors listed.
  2. Response: The authors agree that the addition of gene names for the virulence factors listed in table 1 would improve the manuscript.
  3. Lines amended in the document: Page 5 line 211, Table 1 has been amended with the gene names.

D: Amended text: Gene names have been added for the virulence factors listed in Table 1.

Reviewer 2 Report

Comments and Suggestions for Authors

This is an interesting perspective on an old problem in medical microbiology, the definition and identification of virulence factors. I fully agree with the authors' opinions and I applaud the enthusiasm to try solving this conundrum.

I offer additional points that may be taken on board for manuscript improvement.

The separation of virulence factors from virulence determinants would be relevant. In most cases, the latter are wrongly classified as virulence factors, generating bias in the field.

In the same sense, defining and differentiating pathogenesis factors from virulence factors would be relevant. This is more an anthropocentric view but helps to define whether those putative virulence factors may be found in non. pathogenic species or not.

Finally, I think it should be important to keep in perspective that the interaction of bacteria is not exclusively with humans and other mammals. They interact with fungi, protozoa, plants, and invertebrates. The ability to cause damage is not a generalized trait of bacteria, a species can be harmful to humans but innocuous to invertebrates. Thus, the definition of virulence factors has to always keep the host in perspective. Some virulence factors during interaction with humans may not have that role when interacting with other organisms, for example. This relativity of functions should also be acknowledged in the text.

Author Response

Response to Reviewers:

We are greatly appreciative of the reviewers and the editor for their suggestions on how to improve the manuscript.

Reviewer comments are in black font and our responses to those comments, as well as changes to the text, are in blue font. Page and line numbers of amended text reflect locations within the tracked changes version of the revised manuscript.

Reviewer 2

This is an interesting perspective on an old problem in medical microbiology, the definition and identification of virulence factors. I fully agree with the authors' opinions and I applaud the enthusiasm to try solving this conundrum.

I offer additional points that may be taken on board for manuscript improvement.

The separation of virulence factors from virulence determinants would be relevant. In most cases, the latter are wrongly classified as virulence factors, generating bias in the field.

In the same sense, defining and differentiating pathogenesis factors from virulence factors would be relevant. This is more an anthropocentric view but helps to define whether those putative virulence factors may be found in non. pathogenic species or not.

Finally, I think it should be important to keep in perspective that the interaction of bacteria is not exclusively with humans and other mammals. They interact with fungi, protozoa, plants, and invertebrates. The ability to cause damage is not a generalized trait of bacteria, a species can be harmful to humans but innocuous to invertebrates. Thus, the definition of virulence factors has to always keep the host in perspective. Some virulence factors during interaction with humans may not have that role when interacting with other organisms, for example. This relativity of functions should also be acknowledged in the text.

Response: The authors agree with the reviewer’s comments regarding the perspective of bacterial interaction with different hosts. In the manuscript, we address the utility of using protists as a model system for the investigation of bacterial pathogenesis because it will allow us to expand our understanding at the mechanistic level. There would be a great deal to be gained from comparing the activities of a particular bacterium identified as a human pathogen in mammalian, plant, fungal and protozoan hosts.  Line 164 has been added to the manuscript: “It is important to maintain the perspective that bacteria interact with a variety of organisms including protists, fungi, mammals and other prokaryotes.”

Reviewer 3 Report

Comments and Suggestions for Authors

The author provides a comprehensive overview of the research focal points within the field of microbial pathogenesis, which involve identifying the pathogens and mechanisms that cause diseases. Despite the technological advances that facilitate the rapid identification of pathogens, the scientific rigor in searching for and characterizing virulence factors (VFs) is sometimes lacking. Therefore, the article calls for a reexamination of the epistemology of VFs characterization. Additionally, the paper discusses how certain gene products, which are essentially evolved to support bacterial survival in broader environmental contexts, might be prematurely misclassified as VFs in the context of human biology. By providing examples that lead to such misinterpretations or premature attributions, the authors aim to refine the classification of VFs by emphasizing their biological origins.

Major comments:

a. The paper discusses the lack of scientific rigor in VFs research. How can current research methodologies be specifically improved to enhance scientific rigor and reduce biases?

b. Given the diversity and complexity of VFs, the article proposes new criteria for their reclassification. How will these new standards impact the future direction and strategies of VFs research?

Other minor comments/corrections:

1. The article suggests employing natural protist hosts as models of pathogenicity. How does this approach compare to traditional mammalian models?

2. The article mentions that some VFs may exhibit different pathogenic mechanisms in different hosts. What impact does this variability have on the development of vaccines and pharmaceuticals?

3. Sequence homology plays a critical role in identifying VFs, but it can also lead to misclassification. How should this potential for misguidance be mitigated?

4. Line 176: What is the significance of the underline beneath the text "always"?

5. Line 209: The bacterial name should be italicized. This issue also occurs in the table entry "S. aureus binder of IgG.

Author Response

We are greatly appreciative of the reviewers and the editor for their suggestions on how to improve the manuscript.

Reviewer comments are in black font and our responses to those comments, as well as changes to the text, are in blue font. Page and line numbers of amended text reflect locations within the tracked changes version of the revised manuscript.

Reviewer 3

The author provides a comprehensive overview of the research focal points within the field of microbial pathogenesis, which involve identifying the pathogens and mechanisms that cause diseases. Despite the technological advances that facilitate the rapid identification of pathogens, the scientific rigor in searching for and characterizing virulence factors (VFs) is sometimes lacking. Therefore, the article calls for a reexamination of the epistemology of VFs characterization. Additionally, the paper discusses how certain gene products, which are essentially evolved to support bacterial survival in broader environmental contexts, might be prematurely misclassified as VFs in the context of human biology. By providing examples that lead to such misinterpretations or premature attributions, the authors aim to refine the classification of VFs by emphasizing their biological origins.

Major comments:

Comment #1

  1. The paper discusses the lack of scientific rigor in VFs research. How can current research methodologies be specifically improved to enhance scientific rigor and reduce biases?
  2. Response: The authors acknowledge that current research methodologies are sufficient to warrant continued investigation of the role of a putative VF in disease. However, a goal of this review is to encourage awareness that various forms of bias may hasten conclusions that are not fully warranted. We take a philosophy of science approach to raise awareness that various forms of underdetermination can yield multiple plausible explanations for a given observation.
  3. Lines amended in the document: Page 1, lines 31-32

D: Amended text: In the Importance section, the line “and encourage researchers to consider that there can be multiple plausible explanations for a given observation.”

Comment #2

  1. Given the diversity and complexity of VFs, the article proposes new criteria for their reclassification. How will these new standards impact the future direction and strategies of VFs research?
  2. Response: The authors believe that the criteria proposed in the manuscript will direct researchers to consider the evolutionary origins of putative VFs and consider their role in various environments that includes the mammalian host. Such a global approach to VF research will mitigate the impact of anthropocentric bias that forces the field to focus only on putative roles in human disease. Once the hypothesis that a putative VF has a role in disease has been accepted without consideration of other explanations for in vitro observations then the field can be misdirected for years until a new exemplar is realized.
  3. Lines amended in the document: Page 2, lines 77-78

D: Amended text: A global approach to VF research will mitigate the negative impact that various biases can have on the field of microbial pathogenesis.

Other minor comments/corrections:

Comment #3

  1. The article suggests employing natural protist hosts as models of pathogenicity. How does this approach compare to traditional mammalian models?
  2. Response: Protist hosts can serve as an experimental model for the pathoadaptive changes that contribute to the development of bacterial VFs. For example, validation of genes and their regulatory sequences as virulence attributes is supported experimentally if they are expressed in a both protist and mammalian host. The argument can be made that those genes are maintained for a common function of resistance to phagocytic killing. Any sequence changes that occur in the bacterial genes isolated from a mammalian cell line can lead to an investigation of the specific role that the altered protein has in the two different environments. Mammalian cell cultures are traditionally used as models of human disease but they can fail to provide insight into the origin of a VF production. Protists have been used to isolate viable but nonculturable bacteria presumably due to the bacterium’s adaptation to an intracellular existence that is difficult to mimic in synthetic culture.
  3. Lines amended in the document: Page 9, line 370

D: Amended text: Comparison of the gene sequences of a putative VF that is expressed in both protist and mammalian hosts could lend insight into key pathoadaptive changes.

Comment #4

  1. The article mentions that some VFs may exhibit different pathogenic mechanisms in different hosts. What impact does this variability have on the development of vaccines and pharmaceuticals?
  2. Response: The manuscript states “Although there may be VFs that possess the capacity to cause tissue damage irrespective of the host (e.g., the clostridial neurotoxins [41]) the damage-response framework model informs us that commensalism is relative, and that disease causation may be considered an inappropriate response by the mammalian host to the bacterium.” This reference to the damage-response framework is intended to point out the relative nature of microbial pathogenesis including the concept of a VF. Vaccine and therapeutic development in their current state are directed at ameliorating the effects of a VF as a cause of disease. Anti-toxin antibody inducing vaccines are the classic example. However, by recognizing the role that some bacterial VFs have on an individual host’s immune response for example, it may be possible to accommodate customized therapeutic approaches that do not induce deleterious side effects.
  3. Lines amended in the document: Page 4, line 157

D: Amended text: It may be possible one day to accommodate therapeutic approaches that take into con-sideration an individual host’s immune response to infection.

Comment #5

  1. Sequence homology plays a critical role in identifying VFs, but it can also lead to misclassification. How should this potential for misguidance be mitigated?
  2. Response: A key point of the manuscript is that researchers in the field of microbial pathogenesis should be aware of the biases that are introduced in the search for new VFs. Sequence homologies are an important first step and the guidelines that we have proposed should be applied to any putative VFs whether they have been identified in past reports or recently discovered. In the review, we state that “Rigorous science should always be aware of bias and authors should explicitly state limitations of a particular study.”
  3. Lines amended in the document: No amendments

D: Amended text: No amendments

Comment #6

  1. Line 176: What is the significance of the underline beneath the text "always"?
  2. Response: The line “Rigorous science should always be aware of bias and authors should explicitly state limitations of a particular study.” Emphasis is added to this statement to support the point made by this reviewer in comment #5. Sequence homology searches, as well as in vitro models, can be used to support unwarranted hypotheses in any field of science. A goal of this review is to encourage researchers to be aware that there are various forms of bias that can affect premature conclusions. Authors can promote scientific thinking by simply providing a statement in research report that expresses their awareness of alternate explanations for their hypothesis.
  3. Lines amended in the document: No amendments

D: Amended text: No amendments

Comment #7

  1. Line 209: The bacterial name should be italicized. This issue also occurs in the table entry "S. aureus binder of IgG.
  2. Response: The corrections have been made in the revised manuscript.
  3. Lines amended in the document: Page 5, line 216

D: Amended text: S. aureus has been italicized in Table 1